# Psychometric Properties of the Sibling Attachment Inventory in Mexican Young Adults

**DOI:** 10.3390/ijerph19148570

**Published:** 2022-07-14

**Authors:** Maricela Osorio Guzmán, Massimiliano Sommantico, Carlos Prado Romero, Barbara De Rosa, Santa Parrello

**Affiliations:** 1Facultad de Estudios Superiores Iztacala, Universidad Nacional Autónoma de México, Av. de los Barrios 1, Tlalnepantla de Baz 54090, Mexico; mar1814@yahoo.com (M.O.G.); carlosprador9318@gmail.com (C.P.R.); 2Department of Humanities, University of Naples Federico II, Via Porta di Massa 1, 80133 Naples, Italy; baderosa@unina.it (B.D.R.); parrello@unina.it (S.P.)

**Keywords:** sibling attachment, sibling relationships, satisfaction with life, self-esteem, young adults

## Abstract

The aim of this work was to adapt and validate the Sibling Attachment Inventory (SAI) in Mexican young adults and analyze its psychometric properties. Using an Internet-based survey, data were collected from 307 (64.5% female) Mexican young adults university students (aged 18–30). Exploratory and confirmatory factor analyses were performed to determine the factor structure of the revised Mexican version of the Sibling Attachment Inventory (SAI-RMx). Convergent and predictive validity were verified by carrying out correlations with the parent form of the Inventory of Parent and Peer Attachment (IPPA), the Lifespan Sibling Relationship Scale (LSRS), the Satisfaction With Life Scale (SWLS), and the Rosenberg Self-Esteem Scale (RSE). Results indicated that the SAI-RMx presents good levels of internal consistency and a monodimensional structure, also providing evidence for convergent, predictive, and construct validity. Furthermore, secure attachments were linked with high levels of self-esteem, as well as with high levels of life satisfaction. The SAI-RMx is expected to be a reliable instrument for measuring the global level and the three components of secure attachment between siblings in the young adult’s Mexican population.

## 1. Introduction

The study of affective relationships in recent decades has increased. One of the most solid approaches is the attachment theory [1] which has allowed addressing how people establish their affective relationships to achieve adaptation to their environment [2]. Moreover, this theory allows for analyzing the roles played by parents, siblings, and later peers and romantic partners [3,4].

This is especially relevant since it has been found that the perception of affective attachment relationships is closely related to the psychosocial development of individuals [5], social adjustment [6,7], affective family relationships [8], as well as to the prevention of psychiatric symptomatologies, mood disorders, and antisocial behaviors [9,10]. According to Pardo and colleagues [11], these investigations present empirical evidence of the value of primary attachment and its role as a protective factor against the development of problems in later stages of the life cycle.

One type of bond that is beginning to be studied is the attachment between siblings of young adults which, according to several authors [12,13,14], represents one of the longest, most enduring, and influential relationships that develop across the lifespan. A sibling may also represent one of the earliest attachment figures influencing early psychosocial development a bond that generally lasts a lifetime [15,16,17,18,19,20].

Furthermore, authors such as Whiteman and colleagues [21] report that siblings play a central and permanent role in the lives of individuals and families. In particular, siblings serve as companions, confidants, and role models [22,23] as sources of emotional support and security in the face of various conflicts [24,25,26,27], thus playing an important role as a source of attachment [28].

Moreover, starting from the seminal work of Cicirelli [29], several authors [30,31,32] raise the need to investigate the changes that occur in sibling relationships during the transition from youth to adulthood (18–35 years), since this bond is contingent on the evolution of individuals, specifically in the reorientation of their life projects towards outside the family or new developmental tasks; whether with friends, romantic partners, the culmination of the school stage, entering the world of work, by getting married and by having children [16].

Outstanding data found in different articles indicate that during the transition to adulthood: (a) sibling bonds are characterized by being less conflictive and closer [33]; (b) sibling rivalry for parental attention progressively decreases [34]; (c) and that sibling bond becomes mostly symmetrical, where the age difference between them becomes less relevant, i.e., there is a more equal relationship [35]. It is therefore considered that sibling bonds may act at this stage of the life course as a protective factor for well-being and life satisfaction [36].

Other studies have found that the composition of the sibling dyad influences the type of bond that siblings develop [20]. For example, sister-sibling pairs are characterized by a higher quality of attachment [24], but sister-sibling pairs are defined as less conflictual [37].

Therefore, the adaptation and validation of instruments that evaluate the characteristics of young adults’ sibling bonds represents one of the first steps for psychological research to be developed on the topic and for the results to strengthen the theory.

Because attachment is a construct that cannot be directly observed, that is, it must be inferred from the behaviors and responses of the individual [38], different instruments have been designed to account for this construct; one of them is the Sibling Attachment Inventory (SAI) [25] which is an adaptation of the Inventory of Parents and Peers Attachment (IPPA) [39,40], obtained by substituting the word peer, for that of brother or sister, from which Noel and colleagues [25] hypothesized a similar, although not identical, structure to that of the inventory from which it derives. Indeed, they found that the three factors of the IPPA (Communication, Trust, and Alienation) show a different composition of items for the SAI. Like the IPPA for attachment to parents and peers, the SAI allows for the collection of information on a continuum of attachment to siblings ranging from a “low level of attachment” to a “high level of attachment”. A “high level” of Sibling Attachment describes a person characterized by a positive perception of sibling bonding in terms of good Communication (extent and quality of spoken communication) and Trust (agreement of mutual understanding and respect), as well as of low levels of Alienation (feelings of anger and interpersonal alienation).

Thus, and not having found similar studies in Mexico, this work aimed to adapt and validate the Sibling Attachment Inventory (SAI) in Mexican young adults and to analyze its psychometric properties.

Specifically, and following the procedure performed in a previous study [31], we wanted to verify the factor structure of the revised Mexican version of the Sibling Attachment Inventory (SAI-RMx), as well as to provide evidence of convergent and predictive validity, and verify the reliability of its subscales.

## 2. Method

### 2.1. Participants

Participants were recruited from Mexican universities (websites and students’ mailing lists) from August to November 2019. To enhance this community-based sampling, snowball sampling was also used: participants initially recruited were asked to indicate other potential respondents from their university network. There was no financial reward for participating in the study.

The final sample comprised 307 Mexican university students (64.5% female) aged 18 to 30 years (M = 22.4 years, SD = 3.2). Most participants (49.5%) had one sibling and 45% were first-born (36.5% were second-born, and 18.5% were third or subsequent children). Overall, the siblings identified as preferred were 60.3% male and were aged 7 to 44 years (M = 22.54 years; SD = 7). Most of the participants’ parents (64.8%) were married and the rest were separated or divorced.

### 2.2. Procedure

The study complied with the ethical standards of the American Psychological Association in the treatment of human research participants and was in accordance with the provisions of the 1995 Declaration of Helsinki and subsequent modifications, as well as with the provisions of the Ethical Code of the Psychologist of the Mexican Society of Psychology. The study was reviewed and approved by the Psychological Research Ethics Committee of the Department of Humanities of the University of Naples Federico II (prot. no. 5/2019).

The applied version of the Sibling Attachment Inventory (SAI) [25], was obtained using the same procedure carried out by Parrello and colleagues [31] who performed a translation-back-translation, according to the recommendations of the literature on cross-cultural adaptation of assessment instruments [41].

Data were collected through a self-report questionnaire using a web-based survey [42] comprising: (a) a basic demographic questionnaire; (b) the Spanish adaptation of the Parent form of the Inventory of Parents and Peers Attachment (IPPA); (c) the Sibling Attachment Inventory (SAI); (d) the Lifespan Sibling Relationship Scale (LSRS); (e) the Spanish adaptation of the Satisfaction With Life Scale (SWLS); and (f) the Spanish adaptation of the Rosenberg Self-Esteem Scale (RSE). (See the following paragraph for a detailed description of the measures).

Participation in the study was voluntary and anonymous. Participants were also encouraged to answer by referring to the sibling with whom they have the closest bond. Participants signed an informed consent form that included detailed information about the objectives, study procedure, confidentiality, and anonymity of responses. Participants were instructed to respond by referring to the sibling with whom they had the closest relationship. The survey took approximately 20 min to complete.

### 2.3. Measures

#### 2.3.1. Basic Demographic Questionnaire

A basic demographic questionnaire was created to collect information regarding: (a) age; (b) gender; (c) number of siblings; (d) birth order; and (e) parents’ marital status. Participants were also asked to report the age and gender of the sibling with whom they had the most significant relationship.

#### 2.3.2. Parent Form of the Inventory of Parents and Peers Attachment (IPPA)

The parent form of the IPPA [39,40,43], is a 28-item self-report assessing the attachment on three subscales: Communication (8 items), Trust (10 items), and Alienation (7 items). Each item is rated on a 5-point scale ranging from 1 (Never true) to 5 (Always true). Examples of item are: “I tell my parents about my problems and troubles”, “My parents listen to my opinions”, and “I don’t get much attention at home.” Authors reported satisfactory internal consistency [40]. In the present study, Cronbach’s α were: Communication = 0.86; Trust = 0.88; Alienation = 0.85; and Secure Attachment = 0.87.

#### 2.3.3. Sibling Attachment Inventory (SAI)

The SAI [25] is a 25-item self-report instrument adapted from the IPPA Peer form, by replacing the word peer for the phrase brother or sister. The three subscales evaluating sibling attachment are: (a) Communication (8 items); (b) Trust (10 items); and (c) Alienation (7 items). Each item is rated on a 5-point scale ranging from 1 (Never true) to 5 (Always true). Examples of item are: “When we talk, my brother or sister listens to my opinion”, “My brother or sister accepts me as I am”, and “My brother or sister doesn’t understand my problems.” The Attachment Security score is calculated by summing the scores to the Communication and Trust subscales and subtracting the score to the Alienation subscale.

#### 2.3.4. Lifespan Sibling Relationship Scale (LSRS)

The LSRS [44] is a 48-item self-report instrument that assesses feelings, behaviors, and thoughts related to sibling relationships in childhood and adulthood on six subscales, each composed of 8 items: Child Affect, Child Behavior, Child Cognition, Adult Affect, Adult Behavior, and Adult Cognition. Each item is rated on a 5-point scale ranging from 1 (Strongly disagree) to 5 (Strongly agree). Examples of item are: “My sibling and I ‘hang out’ together”, “I was proud of my sibling when I was a child”, and “My sibling and I have a lot in common.” The LSRS was traduced in Spanish, according to Brislin’s [45] recommendations. Authors of the Italian adaptation and validation [14] reported good internal consistency for each subscale and the total score. In the present study, Cronbach’s α ranged from 0.80 to 0.91 for subscales and was 0.96 for the LSRS total score.

#### 2.3.5. Satisfaction with Life Scale (SWLS)

The SWLS [46,47] is a 5-item self-report instrument assessing global life satisfaction. Each item is rated on a 5-point scale ranging from 1 (Strongly disagree) to 5 (Strongly agree). Examples of item are: “In most ways my life is close to my ideal” and “I am satisfied with my life.” Authors of the Mexican version [47] reported a Cronbach’s α of 0.83. In the present study, Cronbach’s α was 0.86.

#### 2.3.6. Rosenberg Self-Esteem Scale (RSE)

The RSE [48,49] is a 10-item self-report instrument assessing global self-esteem. Each item is rated on a 4-point scale ranging from 0 (Strongly disagree) to 3 (Strongly agree). Examples of item are: “I feel that I have a number of good qualities” and “All in all, I am inclined to feel that I am a failure.” Authors of the Mexican version [49] reported good psychometric properties. In the present study, Cronbach’s α was 0.83.

### 2.4. Data Analyses

First, the database was cleaned, as indicated by Streiner and colleagues [50]. Then, survey data were entered into the Mplus 7.2 [51] and SPSS 26.0 [52].

Exploratory Factor Analysis (EFA; with principal axis factoring and Promax rotation) and Confirmatory Factor Analysis (CFA; with maximum likelihood estimation method) were performed to verify the factor structure of the SAI-RMx. In performing the EFA, factor loadings > 0.55 were considered [53]. Furthermore, in performing the CFA, the following fit indices were calculated: chi-square distribution and degrees of freedom, comparative fit index, Tucker and Lewis Index, root mean squared residual approximation error, standardized root mean square residual, Akaike’s information criterion, and Bayesian information criterion (*χ*^2^*/df* = from 2 to 5; CFI > 0.90; TLI > 0.90; RMSEA = good < 0.05, reasonable < 0.08, and fair < 0.1; SRMR < 0.09; AIC and BIC = smaller values indicate better fit) [54,55,56,57,58]. For reliability analysis, Cronbach’s α was calculated (> 0.70) [59].

The convergent and predictive validity was verified through Pearson’s correlation analysis (*p*-value < 0.05).

Two-way ANOVA (*p* < 0.05) was performed to verify group differences and eta-squared (*η*^2^; small ≥ 0.01; medium ≥ 0.059; large ≥ 0.138) [60] was used to measure effect sizes.

## 3. Results

### 3.1. Factor Structure and Internal Consistency

A preliminary EFA (with principal axis factoring and Promax rotation) was carried out to test the factor structure of the SAI-RMx. The Kaiser–Meyer–Olkin index was 0.961, thus indicating the suitability of the factor analysis. Three components with eigenvalue ≥ 1 were individuated, with the first component accounting for 54.06% of the variance and the other two dimensions accounting for, respectively, 5.2% and 2.6% of the variance.

Through the visual inspection of the scree plot, it was possible to identify a mono-dimensional structure for SAI-RMx, thus indicating the need to force the extraction to a single factor. The resulting solution showed an acceptable level of accounted variance (50.56%) and substantial loadings (*λ* > 0.45), with all items presenting the expected signs on the extracted factor (see Table 1). Only items 10 and 22, belonging to the Alienation subscale, did not present satisfactory loadings.

In line with several studies conducted on the IPPA, our results also justify the use of the global attachment security score for the SAI-RMx. However, as indicated by Parrello and colleagues [31], our results do not exclude the possibility that the subdivision of the attachment construct into two [61,62] or three interrelated dimensions [11,39,43,62,63,64,65,66] can better explain the correlations between the items of the single-factor structure.

Therefore, a CFA and a comparison of the three models were carried out, eliminating items 10 and 22 that presented low loadings (λ < 0.45) in the EFA.

The results show that the fit of the three-factor model was substantially better than the one-factor and two-factor models. Moreover, Cronbach’s α internal consistency indexes were calculated for the three-factor model, which were adequate (see Table 2).

### 3.2. Correlations and Validity

The mono-dimensional structure of the SAI-RMx was highlighted by the high correlation values between the latent dimensions of the three-factor model (*p* < 0.01). Zero-order correlations between the SAI-RMx and the other instruments are shown in Table 3.

### 3.3. Relations between the SAI-RMx, the Parent Form of the IPPA, the LSRS, the SWLS, and the RSE

Regarding the one-factor model, our results indicate significant positive correlations between Sibling Attachment Security and Communication, Trust, and Attachment Security factors of the IPPA parent form (*r* ranging from 0.33 to 0.37; *p* ≤ 0.01) and a significant negative correlation between Sibling Attachment Security and Alienation factor of the IPPA parent form (*r* = −0.32; *p* ≤ 0.01). Results also indicated that Sibling Attachment Security was significantly positively correlated with all LSRS subscales and the total score (*r* ranging from 0.38 to 0.84; *p* ≤ 0.01). Furthermore, results indicate a significant positive correlation between Sibling Attachment Security and SWLS score (*r* = 0.17; *p* ≤ 0.01). Finally, results indicate a significant positive correlation between Sibling Attachment Security and RSE score (*r* = 0.28; *p* ≤ 0.01).

Regarding the three-factor model, results indicate: significant positive correlations between Sibling Communication and Communication, Trust, and Attachment Security factors of the IPPA parent form (*r* ranging from 0.28 to 0.33; *p* ≤ 0.01) and a significant negative correlation between Sibling and Alienation factor of the IPPA parent form (*r* = −0.21; *p* ≤ 0.01); significant positive correlations between Sibling Trust and Communication, Trust, and Attachment Security factors of the IPPA parent form (*r* ranging from 0.28 to 0.36; *p* ≤ 0.01) and a significant negative correlation between Sibling Trust and Alienation factor of the IPPA parent form (*r* = −0.26; *p* ≤ 0.01); significant negative correlations between Sibling Alienation and Communication, Trust, and Secure Attachment factors of the IPPA parent form (*r* ranging from −0.39 to −0.25; *p* ≤ 0.01) and a significant positive correlation between Sibling Alienation and Alienation factor of the IPPA parent form (*r* = 0.48; *p* ≤ 0.01). These findings indicate a medium relationship between the construct of attachment to parents and the construct of attachment to siblings. Indeed, the two constructs were found to be correlated but not totally overlapping. These findings are in line with those obtained from studies that compared the two forms of IPPA (Parent and Peer) [67,68], as well as from studies that compared SAI with the Parent form of IPPA [25,31]. Results also indicate significant positive correlations between all SAI-RMX subscales and LSRS subscales and total score (*r* ranging from −0.58 to 0.84; *p* ≤ 0.01,). Furthermore, results indicate a significant positive correlation between Sibling Trust and SWLS score (*r* = 0.14; *p* ≤ 0.05), as well as a significant negative correlation between Sibling Alienation and SWLS score (*r* = −0.27; *p* ≤ 0.01). Finally, results indicate significant positive correlations between Sibling Communication and Trust and RSE score (*r* respectively 0.18 and 0.25; *p* ≤ 0.01), as well as a significant negative correlation between Sibling Alienation and RSE score (*r* = −0.39; *p* ≤ 0.01). Taken together, these findings corroborate the criterion validity of the SAI-RMx.

### 3.4. Descriptive Statistics and Groups Differences

Descriptive statistics are presented in Table 4. The mean for Attachment Security was sufficiently far from the extremes, as well as the standard deviations were large enough. To verify the answers’ variability, skewness and kurtosis were calculated and have acceptable values.

The main effects of participants’ age, participants’ gender, as well as their interaction on Attachment Security was verified by performing a two-way ANOVA. No significant effects were found (age *F*_3,303_ = 1.481, *p* = 0.13; gender. *F*_1,305_ = 1.666, *p* = 0.54; age by gender *F*_3,303_ < 1). In verifying the main effects of the participants’ age and gender, as well as their interactions, on the Trust and Alienation subscales, similar results were obtained. By the contrary, only a small significant effect of gender was found on the Communication subscale (*F*_1,305_ = 4.581, *p* < 0.05, *η*^2^ = 0.02), with Tuckey post-hoc comparisons showing that females reported significantly higher scores than males (*M_F_* = 27.6, *SD* = 8.22; *M_M_* = 25.5, *SD* = 7.72).

The main effects of sibling’s age and sibling’s gender, as well as their interactions on Attachment Security, were also verified by performing a two-way ANOVA. A small significant effect of gender was only found (*F*_1,305_ = 4.602, *p* < 0.05, *η*^2^ = 0.02), with Tuckey post-hoc comparisons showing that participants whose closest bond was with a brother scored significantly higher than participants whose closest bond was with a sister (*M_F_* = 17.2, *SD* = 3.73; *M_M_* = 16.9, *SD* = 4.51). In verifying the main effects of the sibling’s age and gender, as well as their interactions on the Communication and Trust subscales, similar results were obtained.

Results of the ANOVA and the Tukey post-hoc comparisons indicated that third-born (or later) reported significantly higher scores than others on the Communication subscale (*F*_3,303_ = 3.696, *p* < 0.05, *η*^2^ = 0.02) (*M_I_* = 26.2, *SD* = 7.94; *M_II_* = 26.2, *SD* = 8.31; *M_III_* = 29.4, *SD* = 7.65), on the Trust subscale (*F*_3,303_ = 3.273, *p* < 0.05, *η*^2^ = 0.02) (*M_I_* = 38.9, *SD* = 8.75; *M_II_* = 38.9, *SD* = 8.80; *M_III_* = 42.1, *SD* = 7.19), and on Attachment Security (*F*_3,303_ = 3.604, *p* < 0.05, *η*^2^ = 0.02) (*M_I_* = 48, *SD* = 19.03; *M_II_* = 47.7, *SD* = 19.42; *M_III_* = 55.3, *SD* = 16.97).

Results of the ANOVA and the Tukey post-hoc comparisons also indicated sister–sister pairs showed significantly higher scores than others on the Communication subscale (*F*_3,303_ = 4.300, *p* < 0.01, *η*^2^ = 0.04) (*M_BB_* = 24.4, *SD* = 7.66; *M_BS_* = 28.4, *SD* = 7.62; *M_SB_* = 26.6, *SD* = 8.43; *M_SS_* = 28.6, *SD* = 7.78), while brother-brother pairs reported significantly higher scores than other sibling pairs on the Trust subscale (*F*_3,303_ = 2.797, *p* < 0.05, *η*^2^ = 0.03) (*M_BB_* = 37.6, *SD* = 7.80; *M_BS_* = 42.2, *SD* = 6.72; *M_SB_* = 39.3, *SD* = 9.51; *M_SS_* = 40.4, *SD* = 8.36) and on Attachment Security (*F*_3,303_ = 2.955, *p* < 0.05, *η*^2^ = 0.03) (*M_BB_* = 44.7, *SD* = 17.24; *M_BS_* = 54.6, *SD* = 15.56; *M_SB_* = 48.8, *SD* = 21.33; *M_SS_* = 51.8, *SD* = 17.89).

Finally, regarding number of siblings, siblings’ age difference, and parents’ marital status, no significant differences were found.

## 4. Discussion

Regarding the dimensional structure of SAI-RMx, and in line with the results of Noel and colleagues [25], the CFA showed that the model with the best data fit was the three-factor model. This model is conceptually equivalent to a hierarchical model, constituted by three first-order factors functionally dependent on a second-order factor, thus supporting both the use of the global Attachment Security score and the subscales (Communication, Trust, and Alienation) scores. However, as in previous studies [25,31], high associations were observed between the subscales of the SAI, indicating poor differentiation among the constructs. It is therefore possible to have doubts about the practical use of the three subscales of the SAI.

This situation is not exclusive to the Mexican sample. Indeed, in international research on the factor structure of the IPPA, as well as of the SAI [25,31,67,69], the three subscales were strongly correlated. In addition, there were high factor weights of a single item on different scales, indicating that the content of the items is not clearly referable to the dimension to which it belongs.

The results also support the convergent and predictive validity of the instrument since the SAI subscales are associated with the IPPA and LSRS subscales. Indeed, high scores on the SAI indicate greater closeness and satisfaction in sibling relationships. Similarly, it is demonstrated from the correlations obtained with the SWLS and the RSE that high levels of Attachment Security are linked to high levels of self-esteem and life satisfaction [64,67,69,70].

Discrepancies were found with the study of Noel and colleagues [25], as participant gender, specifically female gender, had a small effect on Communication. In the present study, sibling pairs’ composition had significant effects, however small, on Communication, Communication-Trust, and Attachment Security, data corroborated in previous research [24,71]. In particular, and consistent with authors who describe that sister couples present higher levels of intimacy and warmth [14,72,73], sister-sister pairs scored higher than other sibling pairs on the Communication subscale. On the contrary, and according to Noel and colleagues [25], brother-brother pairs reported significantly higher scores on Trust and Attachment Security.

Furthermore, results indicated younger siblings (third-born or later) showed higher scores on Communication, Trust, and Attachment Security. Concerning these findings, it is possible to hypothesize that younger siblings (third siblings or later) may rely on a more extensive and complex network of attachment figures, consisting of not only parents but also older siblings. These data can also be interpreted in line with the hypothesis of Fraley and Tancredy [74] who note that younger siblings tend to view their older siblings as attachment figures. Moreover, these findings are also in line with Jensen and colleagues [75] who indicated that “younger siblings are more focused on modeling and comparing themselves to their older sibling” (p. 393), thus confirming their significant role as attachment figures played for their younger siblings.

As for Noel and colleagues’ investigation [25], also in the present study siblings’ number had no significant effect on SAI scores.

In contrast with several studies that have found positive associations between parental divorce and insecure attachment in adolescence and young adults [76,77], in the Mexican sample parents’ marital status had no significant effect on sibling attachment. It is likely that, according to previous interpretations [31,78], married families may also promote insecure attachment patterns, while the reconstituted family, following parents’ remarriage, may promote secure attachment patterns. This finding could also reflect the composition of the sample concerning parents’ marital status (64.8% of participants’ parents were married), thus indicating a cultural specificity of the Mexican context, which is characterized by strong religiosity and massive investment in family values [79].

### Strengths and Limitations

To our knowledge, this is the first study to make a useful contribution to the adaptation and validation to the specific Mexican socioeconomic context of a new self-report instrument for the assessment of sibling attachment in young adulthood.

Among the limitations found in this study is the sampling strategy used, our sample only including university students. Furthermore, the snowball technique implies specific possible biases, such as biases related to special characteristics of individuals who voluntarily participate in a study. In this vein, future research could implement other sampling strategies, which would allow the generalizability of results.

Another possible bias in the study was the mono-method one, related to the fact that having assessed all variables of the study by using self-report instruments, there could be inflation in observed associations, as well as missing other observations. In this vein, future research could integrate quantitative data with qualitative data, such as in-depth interviews, to deepen different aspects of sibling relationships. Moreover, the fact of using several additional scales to the SAI can generate respondents’ fatigue and a probable carry-over effect.

Finally, for future research, it is recommended to analyze the ambiguous items considering only those that clearly differentiate between the constructs that make up secure attachment, and it is also proposed to delve into the invariance of the factorial structure between men and women, as well as cross-culturally.

## 5. Conclusions

This study aimed to adapt and validate the Sibling Attachment Inventory to the Mexican context and to analyze its psychometric properties. The results presented above permit us to conclude that this instrument has adequate psychometric characteristics to be used in similar samples. Furthermore, also based on discussed results, we can affirm that the SAI is a reliable measure for measuring Sibling Attachment Security, as well as its components (Communication, Trust, and Alienation).

In this vein, this instrument can be an important tool for the evaluation and study of attachment relationships, specifically between siblings. Research related to attachment theory has often neglected the important role of siblings in this topic, by primarily focusing on parents, peers, and romantic partners.

In addition, the results of the present study may be useful for various investigations to assess validly and reliably the degree to which people take their siblings as attachment figures and to analyze the perceptions that young adults have of their attachment bonds.

Finally, concerning the assessment of the three sub-dimensions originally proposed by Noel and colleagues [25], our results indicate the need for further research aimed at reducing high correlations among the three subscales (Communication, Trust, and Alienation) through the identification of items more specifically related to the constructs to which they belong.

## Figures and Tables

**Table 1 ijerph-19-08570-t001:** Exploratory Factor Analysis Results (*N* = 307).

Item (Subscale)	Factor Loadings	Communalities
6 (T)	0.843	0.710
17 (C)	0.836	0.700
19 (T)	0.836	0.699
20 (T)	0.820	0.672
7 (C)	0.817	0.668
16 (T)	0.810	0.656
2 (C)	0.792	0.628
25 (C)	0.786	0.618
13 (C)	0.780	0.608
12 (T)	0.778	0.605
21 (T)	0.770	0.593
24 (C)	0.768	0.591
15 (T)	0.765	0.585
1 (C)	0.743	0.552
14 (C)	0.729	0.532
8 (T)	0.710	0.504
3 (C)	0.701	0.492
5 (T) (r)	0.682	0.465
11 (A)	−0.644	0.415
4 (A)	−0.609	0.371
23 (A)	−0.500	0.250
9 (A)	0.480	0.230
18 (A)	−0.461	0.212
22 (A)	−0.417	0.174
10 (A)	−0.334	0.112
Eigenvalue 12.25 % of variance 50.56% Cronbach’s *α* 0.88		

*Note:* T = Trust; C = Communication; r = reverse item; A = Alienation.

**Table 2 ijerph-19-08570-t002:** Confirmatory Factor Analyses Results (*N* = 307).

Model	*χ* ^2^	*df*	*χ* ^2^ */df*	CFI	TLI	RMSEA (95% CI)	SRMR	AIC	BIC
One-factor	987.013 *	230	4.29	0.861	0.847	0.104 (0.097–0.110)	0.05	16,953.617	17,210.770
Two-factor	744.250 *	229	3.25	0.913	0.907	0.087 (0.083–0.091)	0.03	15,950.556	16,211.435
Three-Factor	619.710 *	227	2.73	0.980	0.970	0.064 (0.059–0.066)	0.02	15,455.544	16,123.877
Factor *α*									
Attachment Security *α* = 0.87; Com-Tru *α* = 0.96; Communication *α* = 0.94; Alienation *α* = 0.77									

* *p* < 0.001.

**Table 3 ijerph-19-08570-t003:** Zero-order Correlations between the SAI, the IPPA (Parent Form), the LSRS, the SWLS, and the RSE (*N* = 307).

	1	2	3	4	5	6	7	8	9	10	11	12	13	14	15	16	17	18	19
1.C P	1																		
2.T P	0.71 **	1																	
3.A P	−0.65 **	−0.74 **	1																
4.C-TP	0.93 **	0.92 **	−0.75 **	1															
5.A-S P	0.89 **	0.91 **	−0.88 **	0.98 **	1														
6.C S	0.33 **	0.28 **	−0.21 **	0.33 **	0.31 **	1													
7.T S	0.28 **	0.36 **	−0.26 **	0.35 **	0.34 **	0.88 **	1												
8.A S	−0.25 **	−0.35 **	0.48 **	−0.32 **	−0.39 **	−0.53 **	−0.61 **	1											
9. C-T S	0.32 **	0.34 **	−0.25 **	0.35 **	0.34 **	0.97 **	0.97 **	−0.59 **	1										
10.A-S S	0.33 **	0.36 **	−0.32 **	0.37 **	0.37 **	0.94 **	0.97 **	−0.72 **	0.98 **	1									
11. AA	0.30 **	0.36 **	−0.32 **	0.36 **	0.36 **	0.78 **	0.84 **	−0.58 **	0.83 **	0.84 **	1								
12.AB	0.25 **	0.18 **	−0.16 **	0.23 **	0.22 **	0.78 **	0.72 **	−0.46 **	0.77 **	0.76 **	0.70 **	1							
13.AC	0.27 **	0.30 **	−0.22 **	0.31 **	0.30 **	0.76 **	0.80 **	−0.49 **	0.81 **	0.79 **	0.83 **	0.74 **	1						
14.CA	0.23 **	0.26 **	−0.31 **	0.26 **	0.30 **	0.32 **	0.37 **	−0.32 **	0.36 **	0.38 **	0.43 **	0.32 **	0.41 **	1					
15.CB	0.21 **	0.25 **	−0.22 **	0.25 **	0.26 **	0.39 **	0.38 **	−0.27 **	0.40 **	0.40 **	0.41 **	0.40 **	0.40 **	0.66 **	1				
16.CC	0.25 **	0.30 **	−0.30 **	0.30 **	0.31 **	0.42 **	0.44 **	−0.33 **	0.44 **	0.45 **	0.54 **	0.44 **	0.52 **	0.75 **	0.84 **	1			
17.LSRS	0.32 **	0.34 **	−0.32 **	0.36 **	0.37 **	0.72 **	0.74 **	−0.51 **	0.75 **	0.76 **	0.81 **	0.76 **	0.81 **	0.75 **	0.78 **	0.86 **	1		
18.SWLS	0.37 **	0.46 **	−0.53 **	0.45 **	0.49 **	0.09	0.14 *	−0.27 **	0.12 *	0.17 **	0.20 **	0.12 *	0.17 **	0.19 **	0.15 *	0.22 **	0.22 **	1	
19.RSE	0.36 **	0.45 **	−0.54 **	0.43 **	0.49 **	0.18 **	0.25 **	−0.39 **	0.22 **	0.28 **	0.26 **	0.15 **	0.24 **	0.26 **	0.16 **	0.20 **	0.26 **	0.69 **	1

*Note:* C P = Communication IPPA Parent form; T P = Trust IPPA Parent form; A P = Alienation IPPA Parent form; C-T P = Communication-Trust IPPA Parent form; A-S P = Attachment Security IPPA Parent form; C S = Communication SAI; T S = Trust SAI; A S = Alienation SAI; C-T P = Communication-Trust SAI; A-S S = Attachment Security SAI; AA = Adult Affect LSRS; AB = Adult Behavior LSRS; AC = Adult Cognition LSRS; CA = Child Affect LSRS; CB = Child Behavior LSRS; CC = Child Cognition LSRS; LSRS = LSRS total score; SWLS = SWLS total score; RSE = RSE total score. * *p* ≤ 0.05; ** *p* ≤ 0.01.

**Table 4 ijerph-19-08570-t004:** Descriptive Statistics for the IPPA (Parent Form), the SAI, the LSRS, the SWLS, and the RSE (*N* = 307).

Total sample *N* = 307				
	*M*	*SD*	Skewness	Kurtosis
SAI	49.3	18.97	−0.72	0.21
IPPA	49.3	18.37	−0.34	−0.31
LSRS	3.6	0.65	−0.67	0.76
SWLS	3.6	0.85	−0.59	0.03
RSE	13.3	4.07	0.18	−0.33
Males *N* = 109				
	*M*	*SD*	Skewness	Kurtosis
SAI	47.40	17.14	−0.25	−0.54
IPPA	48.8	15.70	−0.35	−0.43
LSRS	3.5	0.61	−0.15	−0.19
SWLS	3.7	0.75	−0.32	−0.18
RSE	12.8	4.08	0.11	−0.66
Females *N* = 198				
	*M*	*SD*	Skewness	Kurtosis
SAI	50.23	19.87	−0.73	0.55
IPPA	49.5	19.72	−0.35	−0.40
LSRS	3.7	0.68	−0.72	0.62
SWLS	3.6	0.90	−0.65	−0.04
RSE	13.6	4.05	0.23	−0.18

## Data Availability

The data presented in this study are available on request from the corresponding author.

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
