# Peer review of "Psychometric Properties of the Sibling Attachment Inventory in Mexican Young Adults"

_ijerph, 2022, doi:10.3390/ijerph19148570_

Round 1

Reviewer 1 Report

The authors adapted the SAI in Mexican young adults and tried to analyse its psychometric properties. Though these results are surely of interest for future attachment research regarding siblings in Mexico, the paper is poorly written, missing to explain concepts of the SAI/IPPA In the introduction and in the methods, overwhelming with results of correlations and not discussing them properly. I would advise major revision of the paper fixing these issues.

Some details and minor comments:

Introduction. Line 56 ff: you start with a), but do not continue. More important, you state that sibling bonds ar closer in l. 57 and less close in l. 60 – this is contradictory. Also the “despite the above”.

I am missing some more work about sibling attachment and later life adversities/mental disorders. There are only l. 63-65, but they stand quite alone and their meaning is not clear.

I also miss in the introduction why you selected the sibling attachment inventory/IPPA to measure attachment in siblings. There are so many other scales with other factors (like attachment avoidance and anxiety) like the ECR (experiences in close relationships). I do not say that I would have preferred this one; I just miss some more explanations about the IPPA construct of attachment, especially because this is very far away from the Ainsworth/Bowlby attachment constructs you cite in the first paragraph. The factors communication, trust and alienation should be explained further and how they represent attachment, also the global attachment security score which comes up first in the results (should be mentioned in the introduction and methods). Any studies containing IPPA and problems in life/mental disorders?

Then, why choosing LSRS and Rosenberg self esteem for comparison? Any studies with the IPPA and these scales?

The methods are missing explanations how attachment security is calculated.

Did you have the information of the years between the participants and the siblings? This might highly influence attachment pattern.

If there are more siblings – to which siblings do you refer when answering the questionnaire?

Table 3 includes many abbreviations, of which only two are explained in the legend. It gives the impression of randomly correlating everything with everything.

Lines 337ff: This somehow seems to mix up your results with literature instead of stating your results and that others have also found these links. It is confusing. Also the paragraph of line 350: just the point that younger siblings see their older ones as attachment figures does not explain why they should have higher scores on communication etc. The arguments are incomplete.

Line 354: older siblings who?

Line 365: some cultural aspects would be nice – are there none, escpecially about divorce? I do not understand “reconstituted  family” when speaking of divorce?

Limitations: line 372: community-based sample? In the methods, it is university sample? University sample does not permit generalization at all.

Paragraph 376 is somehow weird. Of course self-report is a limitation, but it is not about inflation but about missing other observations. But turning to interviews introduces different measures which might not overlap at all.

Actually, after reading the discussion, I am not convinced at all that the SAI “is a reliable measure for measuring Sibling Attachment Security, as well as its 391 components”. The discussion does not pinpoint these important issues clearly enough.

Author Response

Thank you for your valuable suggestions and careful reading of the manuscript.

  • We revised and reorganized sentences between line 56 and 60 in the Introduction;
  • Since the focus of the manuscript is not the relationship between sibling attachment and adversities/mental disorders in later life, we do not consider it necessary to add further references. The same applies to IPPA;
  • Despite the fact that the attachment construct of the SAI, unlike other instruments such as the ECR, is more distant from the Bowlby/Ainsworth constructs, the SAI is currently the only measure specifically developed to assess sibling attachment. Despite this, we have tried to shortly explain this construct and its dimensions in the introduction;
  • Needing to use an instrument for assessing the convergent validity of the SAI, the LSRS was chosen because, in addition to being an instrument that allows assessment of sibling bonding even in adulthood (our sample consists of young adults), it has already been used in the Italian validation of the SAI, as well as the RSE for divergent validity. Furthermore, satisfaction with life and self-esteem are two major aspects of individuals’ psychological adjustment and well-being and it seemed to us important to evaluate the relationships between them and the sibling bond and attachment. Finally, several cited studies (e.g., Gallarin, & Alonso-Arbiol, 2013; Guarnieri et al., 2010; Kocayörük, 2010) used the RSE together with IPPA;
  • We described in the Measures paragraph how Attachment Security is calculated;
  • Regarding siblings’ age difference, no significant differences were found, and we added this data in the Results paragraph. Furthermore, literature on sibling relationships in young adulthood show that in this life phase age difference between siblings becomes less relevant (Scharf et al., 2005; Stewart et al., 2001);
  • In the Procedure paragraph we clarified that participants were also encouraged to answer by referring to the sibling with whom they have the closest bond;
  • We explained all the abbreviations in Table 3;
  • The reasoning presented in the discussion is precisely the opposite. That is, as indicated, the fact that younger siblings (third-born or later) showed higher scores on Communication, Trust, and Attachment Security, could be explained by hypothesizing that younger siblings may rely on a broader network of attachment figures, including older siblings. in the same vein, we refer to Fraley and Tancredy, as well as to Jensen and colleagues, to state that younger siblings tend to view their older siblings as attachment figures, as well as that younger siblings model and compare themselves to their older sibling, thus confirming their significant role as attachment figures.;
  • We eliminated “who”;
  • We briefly specified some cultural characteristics in explaining the finding related to parents' marital status;
  • We revised the Strengths and Limitations paragraph.

Reviewer 2 Report

The Sibling Attachment Inventory adapted by the Parents and Peer Attachment Inventory is one important questionnaire measuring the affective relationships between siblings. Both inventories contain 3 subscales: Communication, Trust and Alienation.. An Italian version exists and is adapted for Mexican young adults.

The sequence of sections is not conclusive. The chapter 2.2. Procedure contains the abbreviations of the used questionnaires, but the references and description of the inventories is found in section 2.3. Measures – it would be more suitable to change the order of the two sections,  Measures before procedures.

As sample 307 young Mexicans answered the inventories, but only 49.4% had siblings. How the data of the people without siblings was dealt with? How could they estimate their relationship with “siblings”?

The abbreviations of the respective subscales should be explained in notes at the tables – e.g. in the table of the correlations between all measures it is unclear what subscales of what measures are negatively or positively correlated.

Tale 4- why no differences between young Mexicans with and without siblings are given?

Author Response

Reviewer 2

Thank you for your valuable suggestions and careful reading of the manuscript.

  • Normally in articles the Procedure paragraph precedes the Measures paragraph. Thus, instead of reversing the order of the paragraphs, we have taken care to indicate in the Procedures paragraph that the measures will be described in detail in the next paragraph;
  • There was an error in writing. All participants had siblings, and 49.5% had one (and not a) sibling. Precisely because of this, there is no data regarding people without siblings, as well as no reported differences between young Mexicans with and without siblings;
  • We explained in Table 3 all the abbreviations of the respective subscales.

Reviewer 3 Report

Dear authors,

Congratulations on your excellent paper. I have really enjoyed reading it. This is exactly what I expect when I am reviewing a paper that includes exploratory and confirmatory factor analysis of a research instrument. You have conducted and reported very well everything in the exploratory factor analysis, indicating the communalities and how step by stpe you take every decision. Goodness and badness of fit analysis results reported in your manuscript are great. I have rarely seen a CFI of .98 before. brilliant. that three factor model is awesome.

My only suggestion to improve the manuscript is that, before carrying out an exploratory factor analysis, you should indicate if it is suitable or not. I know your results are great and sure it is not necessary, but, in a formal way, you should have done that before carrying out the exploratory analysis. Kaiser-Meyer-Olkin and Bartlett's sphericity test allow you to do that and are available at the SPSS 26.0 software you have used in your research. If you have the chance, please include it in your paper. If not, doesn't matter, I am just going to say it can be accepted in its present form. Congratulations!

Author Response

Reviewer 3

Thank you for your appreciation and careful reading of the manuscript.

  • We reported a Kaiser-Meyer-Olkin index of .961, thus indicating the suitability of the EFA in the second line of the paragraph Factor Structure and Internal Consistency.

Round 2

Reviewer 2 Report

It is ok